# Clinical presentation and 3-year outcomes of patients with acute coronary syndromes and non-obstructive coronary arteries on angiography

**Lukasz Zandecki**[1]*, **Agnieszka Janion-Sadowska**[2], **Jacek Kurzawski**[2], **Lukasz Piatek**[1], **Michal Zabojszcz**[1], **Krzysztof Plens**[3], **Zbigniew Siudak**[1], **Marcin Sadowski**[1]

**1** Collegium Medicum, Jan Kochanowski University, Kielce, Poland, **2** Swietokrzyskie Cardiology Center, Kielce, Poland, **3** Krakow Cardiovascular Research Institute, Krakow, Poland

* lukasz.zandecki@gmail.com

**Data Availability Statement:** The data used in this study is from the ORPKI registry run by Jagiellonian University Medical College (Krakow,

## Abstract

### Background

With the emerging interest in patients with myocardial infarction with non-obstructive coronary arteries (MINOCA), there is a need to define an even broader group of patients with the syndrome of myocardial ischemia with non-obstructive coronary arteries (INOCA). There are limited data on the clinical characteristics and prognoses of such patients who present with symptoms of acute coronary syndrome (ACS) and undergo urgent coronary angiography that reveals no significant lesions. The aim of this observational study was to compare patients with ACS INOCA and those with ACS with obstructive coronary artery disease (OCAD) both within unadjusted cohorts and with propensity score matched controls.

### Methods and results

This observational study was based on the data from the Polish National Registry of Invasive Cardiology Procedures. Of 9744 patients included, 7624 had OCAD and 2120 had ACS INOCA. In unadjusted cohorts, the overall survival and incidence of major adverse cardiovascular events (MACE: death, cardiac arrest, myocardial infarction, stroke, and heart failure hospitalization) until 36 months were higher in patients with ACS OCAD. Following propensity matching, higher win ratios of death (p = 0.02), additional revascularizations by percutaneous coronary intervention or coronary artery bypass graft surgery (p<0.001), and cardiac hospitalization (p<0.001) were observed in these patients. In contrast, the win ratios of myocardial infarction (p = 0.74), heart failure hospitalization (p = 0.86), and MACE (p = 0.07) were not significantly different between the groups.

### Conclusions

The prognosis of patients with ACS INOCA was more favorable than that of patients with ACS OCAD; however, the differences diminished after adjustments for the initial clinical

Poland) and endorsed by the Ministry of Health and Polish Society of Cardiology. Access to data from the registry needs to be applied for and third-party data usage is not allowed, irrespective of whether the data contain potentially identifying or sensitive data or not. Access to ORPKI data supporting the present findings can be applied for from the ORPKI Registry coordinator (zbigniew.siudak@gmail. com). The Data that researchers will be able to access fits the Journal's definition of minimal Data set.

**Funding:** The project is supported under the program of the Minister of Science and Higher Education under the name "Regional Initiative of Excellence" in 2019-2022 project number: 024/RID/2018/19, financing amount: 11.999.000,00 PLN. The funders had no role in study design, data collection and analysis, decision to publish, or preparation of the manuscript.

**Competing interests:** The authors have declared that no competing interests exist.

profiles. An ACS incident should not be judged as trivial even when cardiac markers remain stable and no significant lesions are found on angiography.

## Introduction

More than 10% of patients who present with symptoms suggestive of acute myocardial infarction have no significant atherosclerotic plaques on coronary angiography. This clinical condition has been termed myocardial infarction with non-obstructive coronary arteries (MINOCA) [1]. The diagnosis of MINOCA is established immediately in a patient who presents with features consistent with acute myocardial infarction, non-obstructive coronary arteries on angiography and no clinically specific cause for the acute presentation [2]. Another group of patients present with symptoms suggestive of acute coronary ischemia and undergo urgent coronary angiography despite negative cardiac biomarkers. A majority of them are diagnosed with unstable angina, unless a specific cause for their acute presentation is identified because they do not fulfill the criteria of MINOCA. There are limited epidemiological data available on their clinical characteristics and prognoses. Recently, a new concept has been proposed to define an even broader group of patients with syndrome of myocardial ischemia with non-obstructive coronary arteries (INOCA) [3].

The pathophysiology of MINOCA / INOCA is complex with several possible underlying mechanisms [2,4–6]. Patients with MINOCA are usually younger and more often females; however, the distribution of the traditional cardiovascular risk factors and comorbidities varies between analyses [5,7–9]. Data on clinical profiles of patients with INOCA presenting with ACS are scarce [3,10]. Atherosclerosis is considered an important mediator of both INOCA and MINOCA syndromes [3], and ruptured atherosclerotic plaques appear on coronary imaging in approximately 40% of these patients [11,12]. Most studies focus specifically on patients with MINOCA rather than the whole group of patients with ACS INOCA; the reported annual mortality rate in MINOCA is approximately 2% with significant heterogeneity across studies [13]. The all-cause mortality at 12 months was reported to be lower in patients with MINOCA than in those with myocardial infarction (MI) with obstructive coronary artery disease (OCAD) [5]. However, a recent large study demonstrated that clinical outcomes were similar between patients with MINOCA and those with OCAD MI [14]. Data on patients with ACS INOCA patients are limited but there is emerging evidence that patients with INOCA are at a higher risk of future adverse clinical events, including death, MI, stroke, or heart failure, than was previously believed [10].

The main objective of the study was to assess and compare clinical characteristics and prognoses of patients who presented with symptoms of an acute coronary syndrome (ACS) and underwent urgent coronary angiography, which revealed normal coronary arteries or only non-significant lesions (ACS INOCA group), and to compare them with patients with ACS and OCAD (ACS OCAD group) within unadjusted cohorts and with propensity score matched controls.

## Materials and methods

### Study design and population

This observational study was based on the data from the Ogolnopolski Rejestr Procedur Kardiologii Inwazyjnej (ORPKI registry—Polish National Registry of Invasive Cardiology Procedures). ORPKI is an electronic database operated by the Jagiellonian University Medical

College in Krakow and endorsed by the Polish Association of Cardiovascular Interventions of the Polish Cardiac Society that collects data on all percutaneous procedures in interventional cardiology performed in Poland [7,15]. Data in the ORPKI registry have been gathered via electronic case report forms. Data on patients with a diagnosis of ACS (ST-segment elevation MI, non-ST-segment elevation MI or unstable angina) treated in the Swietokrzyskie region of Poland and included in the ORPKI registry between January 1, 2014, and December 20, 2017, were analyzed in this study. Swietokrzyskie is a region in southeast Poland with over 1.2 million inhabitants. In the region, seven centers with catheterization labs actively participated in the ORPKI initiative and recruited patients throughout the study period. Patients with ACS INOCA were identified based on non-obstructive coronary arteries visualized on angiography. In case any significant lesion (>50%) was found, a diagnosis of ACS OCAD was established. All patients provided informed consent for the procedures. All medical procedures were performed according to the current medical standards. The study complied with the ethical principles for clinical research based on the Declaration of Helsinki with its later amendments.

The patients were followed up for up to 36 months to assess mortality and major adverse cardiovascular events (MACE), which were defined as the composite of death, cardiac arrest, MI, stroke, or heart failure hospitalization. Data regarding all-cause mortality, hospitalization for revascularization, and MACE were obtained from the official records of the National Health Fund.

## Statistical analysis

Categorical variables were presented as percentages. Continuous variables were presented as mean ± standard deviation or median with quartiles (IQR, interquartile range). Normality was assessed using the Kolmogorov–Smirnov–Lilliefors (KSL) test for samples of over 2000 observations or the Shapiro–Wilk test, as appropriate. Equality of variances was assessed using the Levene's test. Differences between groups were compared using the Student's or Welch's t-test based on the equality of variances for normally distributed variables. The Mann–Whitney U test was used for non-normally distributed continuous variables or ordinal variables. Categorical variables were compared using the Pearson's chi-squared test or Fisher's exact test if 20% of cells had expected count <5. The survival function was estimated using the Kaplan–Meier estimator and comparisons between subgroups were performed using the log-rank test.

To avoid the potential influence of non-randomized design and reduce bias, a propensity score was calculated using a multivariate logistic regression model with ACS INOCA diagnosis as the dependent variable and the following variables: site, age, sex, weight, diagnosis, diabetes, previous stroke, previous MI, previous percutaneous coronary intervention (PCI), previous coronary artery bypass graft surgery (CABG), active smoking status, hypertension, kidney disease, cardiac arrest at baseline, access site during angiogram, chronic obstructive pulmonary disease, and Killip class on admission. Covariate balance was assessed using standardized differences that were less than 0.05. Pairs of patients with ACS INOCA and ACS OCAD were formed using 1:1 caliper matching with caliper width of 0.03. Unpaired patients were excluded from the analysis. Continuous parameters were compared for pairs using paired Student's t-test if differences between pairs were normally distributed or using Wilcoxon signed-rank test otherwise. For nominal variables, the McNemar's or Bowker's test was used. For lifetime data, the matched pairs approach of the win ratio method was applied [16]. The level of statistical significance was set at p<0.05. Statistical analyses were performed using JMP® v14.0.0 (SAS Institute INC., Cary, NC, USA) and R v3.4.1 (R Foundation for Statistical Computing, Vienna, Austria, 2017).

## Results

Overall, 9771 patients with ACS were identified. Twenty-seven (0.3%) patients were excluded from the analysis due to missing or inconsistent data, and 9744 patients were included. Of them, 7624 had OCAD and 2120 were eventually diagnosed with ACS INOCA. Baseline characteristics of the patients are summarized in Table 1, and their unadjusted in-hospital outcomes are summarized in Table 2. On average, the patients with INOCA were only slightly

**Table 1. Baseline characteristics of acute coronary syndrome patients with OCAD and INOCA.**

| Patient characteristics | OCAD n = 7624 | INOCA n = 2120 | p value |
|---|---|---|---|
| Age, years | 67.54 (SD 10.93) | 66.58 (SD 10.66) | 0.006 |
| Female sex | 2471 (32.4%) | 1111 (52.4%) | < 0.001 |
| Diagnosis | | | < 0.001 |
| ST-elevation myocardial infarction | 2211 (29.0%) | 38 (1.8%) | |
| Non ST-elevation myocardial infarction | 1352 (17.7%) | 181 (8.5%) | |
| Unstable angina | 4061 (53.3%) | 1901 (89.7%) | |
| Weight, kg | 79.71 (SD 14.41) | 78.88 (SD 14.82) | < 0.001 |
| Diabetes mellitus | 1713 (22.5%) | 292 (13.8%) | < 0.001 |
| Previous stroke | 263 (3.5%) | 58 (2.7%) | 0.1 |
| Previous myocardial infarction | 1516 (19.9%) | 293 (13.8%) | < 0.001 |
| Previous percutaneous coronary intervention | 1241 (16.3%) | 340 (16.0%) | 0.79 |
| Previous coronary artery bypass graft surgery | 347 (4.6%) | 63 (3.0%) | 0.001 |
| Smoking—active | 1638 (21.5%) | 238 (11.2%) | < 0.001 |
| Hypertension | 5382 (70.6%) | 1461 (68.9%) | 0.14 |
| Kidney disease | 325 (4.3%) | 54 (2.6%) | < .001 |
| Chronic obstructive pulmonary disease [a] | 78 (1.5%) | 32 (2.2%) | 0.047 |
| Killip class on admission [a] | | | < 0.001 |
| I | 3982 (91.5%) | 1051 (94.7%) | |
| II | 260 (6.0%) | 48 (4.3%) | |
| III | 55 (1.3%) | 8 (0.7%) | |
| IV | 53 (1.2%) | 3 (0.3%) | |
| Cardiac arrest at baseline | 69 (0.9%) | 5 (0.2%) | 0.002 |
| Access site during angiogram | | | < 0.001 |
| Radial | 4667 (61.3%) | 1513 (71.5%) | |
| Femoral | 2930 (38.5%) | 597 (28.2%) | |
| Other | 16 (0.2%) | 5 (0.2%) | |
| Additional coronary artery assessment (FFR, IVUS, OCT) | 70 (0.9%) | 34 (1.6%) | 0.007 |
| Results of angiography | | | < 0.001 |
| No evidence of atherosclerosis | 0 (0.0%) | 456 (21.5%) | |
| Without significant stenosis | 0 (0.0%) | 1664 (78.5%) | |
| Single vessel disease | 3612 (47.4%) | 0 (0.0%) | |
| Left main disease | 43 (0.6%) | 0 (0.0%) | |
| Multivessel disease | 3186 (41.8%) | 0 (0.0%) | |
| Multivessel and left main disease | 783 (10.3%) | 0 (0.0%) | |
| Myocardial bridge | 8 (0.2%) | 9 (0.9%) | 0.003 |

OCAD: obstructive coronary artery disease, INOCA: myocardial ischemia with no obstructive coronary arteries, SD: standard deviation, FFR: fractional flow reserve, IVUS: intravascular ultrasound, OCT: optical coherence tomography.

[a] data not available for all patients.

**Table 2. In-hospital outcomes of acute coronary syndrome patients with OAD and INOCA (unadjusted and after matching on propensity scores).**

| Outcomes until discharge | Unadjusted patients cohorts | | | Patients matched on propensity scores | | |
|---|---|---|---|---|---|---|
| | OCAD n = 7624 | INOCA n = 2120 | p value | OCAD n = 1936 | INOCA n = 1936 | p value |
| Length of stay in hospital, days | 5 (4–7) | 3 (2–5) | < 0.001 | 5 (3–7) | 3 (2–5) | < 0.001 |
| Death | 289 (3.8%) | 11 (0.5%) | < 0.001 | 40 (2.1%) | 11 (0.6%) | < 0.001 |
| Cardiac arrest | 0 (0.0%) | 0 (0.0%) | - | 0 (0.0%) | 0 (0.0%) | - |
| Myocardial infarction | 1 (0.0%) | 1 (0.1%) | 0.39 | 0 (0.0%) | 1 (0.1%) | - |
| Stroke | 0 (0.0%) | 0 (0.0%) | - | 0 (0.0%) | 0 (0.0%) | - |
| New or worsening HF | 3 (0.0%) | 1 (0.1%) | 1.0 | 1 (0.1%) | 1 (0.1%) | 1 |
| MACE [a] | 293 (3.8%) | 13 (0.6%) | < 0.001 | 41 (2.1%) | 13 (0.7%) | < 0.001 |

HF: heart failure. Other abbreviations as in Table 1.

[a] Major adverse cardiovascular event defined as death, cardiac arrest, myocardial infarction, stroke, new or worsening HF.

younger and more often female. They were less likely to have traditional cardiac risk factors, such as diabetes or active smoking. Patients in the ACS INOCA group had a more favorable Killip class distribution and were significantly less likely to suffer cardiac arrest on admission than those in the ACS OCAD group. A majority of patients in the INOCA group had unstable angina and only 10.3% had MI confirmed by elevated cardiac biomarkers for a diagnosis of MINOCA.

Radial access site for coronary angiography was the commonest site used in both groups and more so in the INOCA group. In patients with ACS OCAD, 6061 (79.5%) had *ad hoc* PCI, 365 (4.8%) had PCI during index hospitalization, 366 (4.8%) underwent CABG during index hospitalization, and 155 (2%) were scheduled for CABG within 30 days. All patients with ACS INOCA were treated conservatively.

The study population was followed up for a median of 543.5 (240–836) days. The frequencies of MACE and mortality were higher in patients with ACS OCAD at discharge and up to the 36-month follow-up (Tables 2 and 3, Fig 1A and 1B). Kaplan–Meier curves for overall survival and MACE until 36 months in the unadjusted ACS INOCA and ACS OCAD groups are presented in Fig 1A and 1B.

**Table 3. Three-year outcomes of acute coronary syndrome patients with OAD and INOCA (unadjusted and after matching on propensity scores).** Only patients with documented event or completed 36-month follow-up are included.

| Outcomes until 36 months | Unadjusted patient cohorts | | | Patients matched on propensity scores | | |
|---|---|---|---|---|---|---|
| | OCAD | INOCA | p value | OCAD | INOCA | p value |
| Death | 1053 (57.5%) | 150 (38.8%) | < 0.001 | 183 (44.1%) | 150 (42.0%) | 0.09 |
| Cardiac arrest | 10 (1.3%) | 4 (1.7%) | 0.75 | 1 (0.4%) | 4 (1.9%) | 0.32 |
| Myocardial infarction | 152 (16.8%) | 31 (11.8%) | 0.0499 | 22 (8.9%) | 29 (12.5%) | 0.48 |
| Stroke | 56 (6.8%) | 5 (2.1%) | 0.006 | 11 (4.6%) | 5 (2.4%) | 1.0 |
| HF hospitalization | 431 (38.0%) | 109 (33.8%) | 0.16 | 97 (31.3%) | 100 (34.9%) | 0.41 |
| MACE [a] | 1578 (70.4%) | 273 (56.9%) | < 0.001 | 291 (58.9%) | 262 (59.1%) | 0.3 |
| PCI | 1862 (75.8%) | 13 (5.2%) | < 0.001 | 425 (71.2%) | 13 (5.9%) | < 0.001 |
| CABG | 701 (50.0%) | 24 (9.3%) | < 0.001 | 218 (50.1%) | 22 (9.8%) | < 0.001 |
| Cardiac hospitalization | 3802 (91.2%) | 459 (74.0%) | < 0.001 | 964 (91.7%) | 435 (76.2%) | < 0.001 |

Abbreviations as in Tables 1 and 2.

[a] Major adverse cardiovascular event defined as death, cardiac arrest, myocardial infarction, stroke, or HF hospitalization.

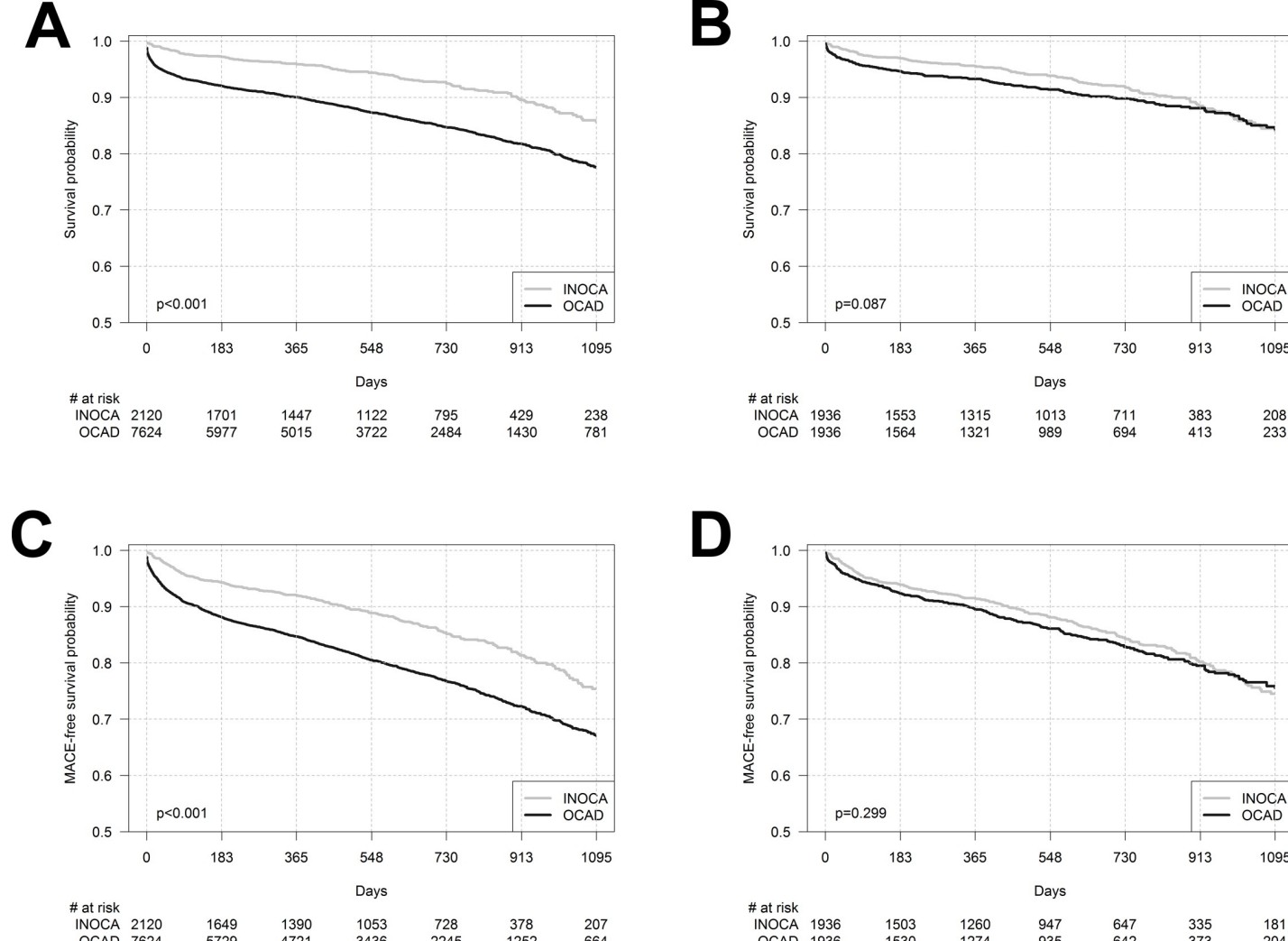

**Fig 1. Kaplan-Maier curves for the following.** A, days-to-death in unadjusted cohorts of patients with ACS INOCA and those with ACS OCAD. B, days-to-MACE (death, cardiac arrest, MI, stroke, and HF hospitalization) in unadjusted cohorts of these patients. C, day-to-death in propensity score matched cohorts of these patients. D, days-to-MACE (death, cardiac arrest, MI, stroke, and HF hospitalization) in propensity score matched cohorts of these patients. ACS, acute coronary syndrome; INOCA, myocardial ischemia with no obstructive coronary arteries; OCAD, obstructive coronary artery disease; MI, myocardial infarction; HF, heart failure.

Outcomes of patients with ACS INOCA in whom no evidence of atherosclerosis was observed on angiography were not significantly different from those in patients in whom non-significant stenosis was observed. During a median follow-up of 585.5 (251–861) days, MACE occurred in 57 (12.5%) and 259 (15.6%) patients, respectively (p = 0.1).

To control for the baseline characteristics, 1936 patients with ACS INOCA were matched with 1936 patients with ACS OCAD using propensity scores. Subsequently, the in-hospital mortality was still higher in the ACS OCAD group. In-hospital outcomes of the two groups are summarized in Table 2. During the follow-up, differences in mortality and MACE between the matched cohorts attenuated over time. Kaplan–Meier curves for overall survival and MACE until 36 months in propensity score matched cohorts of the two groups are presented in Fig 1C and 1D.

The win ratio approach demonstrated a higher probability of death in the ACS OCAD group; however, the statistical significance of this observation was borderline (p = 0.02).

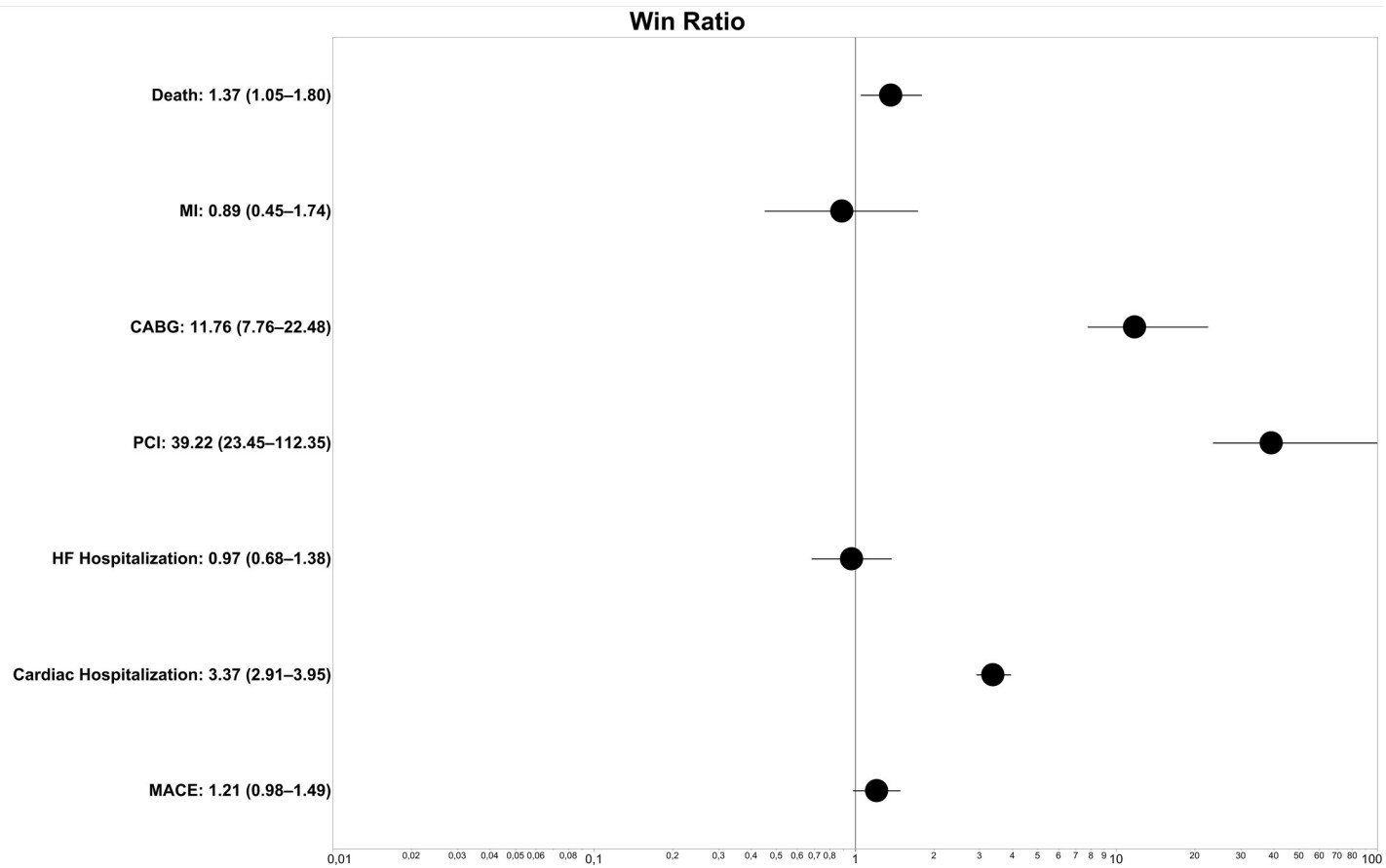

**Win Ratio**

**Fig 2. Forest plot of win ratio (with 95% confidence intervals) for propensity score matched pairs of patients with ACS INOCA and ACS OCAD.** ACS: acute coronary syndrome, INOCA: myocardial ischemia with no obstructive coronary arteries, OCAD: obstructive coronary artery disease, MI: myocardial infarction, HF: heart failure, PCI: percutaneous coronary intervention, CABG: coronary artery bypass graft surgery.

Patients who initially presented with ACS OCAD had a markedly higher ratio of additional revascularization by PCI or CABG (p<0.001) and approximately three times higher ratio of cardiac hospitalization (p<0.001) than in patients who initially presented with ACS INOCA. In contrast, the win ratios for MI and HF hospitalization during the follow-up did not differ between the groups (Fig 2). The win ratio for MACE was not significantly different (p = 0.07) between the groups after propensity score matching (Fig 2).

## Discussion

The main finding of this study was that although the prognosis was generally more favorable in patients with ACS INOCA than in those with ACS OCAD, prognosis appears to be largely related to their cardiovascular risk characteristics and baseline clinical presentations because differences diminished after adjustments for the initial clinical profiles.

### Clinical characteristics of patients with ACS INOCA

There were notable differences in the initial clinical characteristics of patients in the two groups. Patients with ACS INOCA tended to be younger and had a different sex distribution than those with ACS OCAD. The mean age of patients with ACS INOCA was 65.6 years,

which was close to the upper limit reported in previous MINOCA studies [5,17]; however, it was still lower than that in patients with ACS OCAD (67.5 years).

Some studies have demonstrated that the prevalence of traditional cardiac risk factors is lower in patients with MINOCA than in those with MI and OCAD [7,18]. In contrast, among the most common CAD risk factors, a recent systematic review has demonstrated that only hyperlipidemia was significantly less frequent in patients with MINOCA [5]. In our patients with ACS INOCA, traditional cardiovascular risk factors including diabetes, active smoking, previous MI, or kidney disease were significantly less frequent than in those with ACS OCAD. Additionally, their hemodynamic presentation on admission was more favorable; more patients had no heart failure symptoms (Killip class I) and fewer patients presented with cardiogenic shock (Killip class IV) or cardiac arrest at baseline.

The definition of MINOCA includes patients with either normal coronary arteries (no stenosis or stenosis <30%) or mild coronary atherosclerosis (stenosis >30% and <50%) [2]. In our study, the majority of patients with ACS INOCA (78.5%) had at least some evidence of coronary atherosclerosis on angiogram, which was higher than that reported in a recently published meta-analysis of patients with MINOCA in which non-obstructive CAD was present in 54% of patients [13]. This suggests that elevated biomarkers do not necessarily predict the burden of coronary atherosclerosis.

Clinical profiles of patients with ACS INOCA were different from those of patients with ACS in whom OCAD would be subsequently identified; therefore, adjustments were necessary to compare outcomes between the groups that differed only in baseline angiography findings. It is not clear if the risk assessment tools used for predicting adverse clinical outcomes in patients with MI and OCAD should also be used in patients with INOCA [10]. At least they appear to be more useful in predicting the clinical outcomes in patients with MINOCA [17]. Multiple factors have been proposed as independent predictors of adverse outcomes in patients with MINOCA, and most of them are related to patient characteristics on admission [17,19]. These factors were also available for propensity score estimation in this study.

## In-hospital outcomes

Mortality rates reported in patients with OCAD are generally higher than those in patients with MINOCA; however, the difference does not reach statistical significance [18]. To some extent, this may be linked to a relatively low incidence of death and limited sample sizes. Data from a specific population of a "super-aging society" in Japan indicated that the short-term prognosis is worse in patients with MINOCA than in those with MI and CAD [20]. A recent meta-analysis of patients with MINOCA reported in-hospital mortality of 0.9% [5]. In the present study, in-hospital mortality rate of approximately 0.5% was observed in patients with INOCA and was lower than that in patients with ACS OCAD both before and after propensity matching.

## Follow-up mortality and MACE after propensity score matching

After adjusting for baseline risk factors, the prognosis tended to be similar in patients with MI irrespective of the presence of obstructive CAD [14,21]. Our study partially confirms these observations in a broader group of patients with ACS, including those without elevated cardiac biomarkers. Despite better in-hospital outcomes in patients with ACS INOCA, the difference in MACE attenuated over time. In terms of the incidence of MACE during the follow-up, ACS carried a prognosis which was only numerically better (p = 0.07) in patients with INOCA than in those with OCAD after propensity score matching. The win ratio for all-cause death remained higher in patients with OCAD, while that for MI and HF hospitalizations was similar

between the groups. Rather surprisingly, in a recent Chinese study, the incidence of non-fatal MI or HF was extremely low even in patients with MINOCA over a 1-year follow-up [9]. Elevated cardiac biomarkers are known to be associated with worse prognosis in a variety of clinical scenarios. However, ACS presentation that convinces a treating physician to refer the patient for urgent angiography should not be regarded as a trivial incident, even if no significant lesions are identified and cardiac biomarkers remain stable. In a group of patients with ST-elevation ACS triaged for urgent coronary angiography, patients with INOCA had a similar or higher long-term risk of death than those with obstructive CAD, irrespective of the cardiac troponin levels [8].

## The risk of subsequent revascularizations

Patients with ACS INOCA and OCAD are anatomically very different from each other; therefore, the risk of subsequent revascularization during the follow-up period was, as expected, higher in patients with ACS OCAD. This is in accordance with the findings of a previous study that reported an over 6-fold higher risk of any repeat revascularization [14]. The magnitude of this risk was even higher in the current analysis (Fig 2) that included mostly patients with ACS but without a diagnosis of MI. The possible explanation is that CAD could be likely diagnosed at an earlier stage in these patients with ACS INOCA, and one could not determine if they would have undergone revascularizations later after the completion of follow-up. In fact, approximately half of patients with reinfarction after MINOCA who underwent coronary angiography had progression of coronary stenosis [22]. The higher win ratio for any cardiac hospitalization could be explained by the higher rate of repeated revascularizations, whereas the ratios of hospitalizations for HF were not different between the groups.

## Options for secondary prevention

Optimal secondary prevention strategies are not yet established for ACS INOCA, and treatment guidelines do not specifically address this population. MINOCA should be considered as a working diagnosis that requires further diagnostic workup to identify possible coronary or extra-coronary pathologies [2,6,23]. This approach is likely to be true for INOCA as well. However, additional coronary artery assessment is rarely performed in everyday clinical practice [7] or not reported in large registries [22]. This was also true in this study group where only 1.6% of patients with INOCA underwent additional coronary artery assessments while data on any further diagnostic workout outside the catheterization lab were not available. Given the many possible underlying causes of INOCA and lack of systematic evaluation of the underlying mechanisms, treatment recommendations remain empiric [2]. Factors predicting unfavorable outcomes in patients with MINOCA appear similar to factors previously demonstrated to predict new cardiovascular events in patients with OCAD [17]. The comparison of Kaplan–Meier curves for unadjusted and propensity score matched cohorts (Fig 1) suggest the effects of baseline characteristics—which included most traditional CAD risk factors—on the magnitude of differences in outcomes. Management guided by an additional assessments of functional coronary arteries [24] could be the optimal strategy; however, such an approach is currently underutilized. Additionally, myocarditis may account for approximately 33% cases of MINOCA [6]; therefore, additional tests including cardiac magnetic resonance imaging with late gadolinium enhancement could entirely change the management strategy. Nevertheless, we believe that in patients with ACS INOCA in whom no overt diagnosis for ACS background can be established, the management should focus on the known CAD risk factors irrespective of results of angiography because, apart from the rates of subsequent revascularization procedures, patients with ACS INOCA had similar outcomes in terms of subsequent MI

or heart failure hospitalizations over 3 years of follow-up as patients who initially presented with ACS OCAD. This strategy is also supported by the fact that the frequency of non-significant lesions in patients with ACS INOCA in our study (78.5%) was relatively high. However, the 3-year all-cause mortality was still slightly higher in those with ACS OCAD than in those with ACS INOCA. It appears likely that the cardiac risk of patients with INOCA changes over time and the ACS incident should warrant appropriate cardiovascular risk management rather than reassurance of the benign nature of this condition manifesting as negative cardiac biomarkers and non-obstructive coronary arteries on angiography.

## Limitations

Several limitations of our study should be acknowledged. In the ORPKI registry, all data about the patients are entered by the local physicians according to the registry protocol. Therefore, there were some potentially important baseline data missing, especially regarding the laboratory and echocardiographic findings. This may affect the quality of our propensity score model. Importantly, data on the diagnostic workup for possible non-cardiac causes of an acute presentation were missing in the registry. The diagnosis of ACS and referral for coronary angiography was performed in each participating institution according to the current medical standards. Furthermore, additional tests to continue the diagnostic workup of patients with INOCA during and after coronary angiography were limited. This may be an important source of bias because some of the included patients with INOCA might have had non-ischemic or even non-cardiac causes of their acute presentations. However, given the retrospective nature of the registry data, we could only emphasize the problem that the current utilization of these tests in the real world is far from optimal. Furthermore, the ORPKI registry does not include data on pharmacotherapy after discharge, and many patients enrolled late during the study period were followed up for less than 3 years. The study included patients treated in one region in Poland; therefore, caution is warranted before generalizing our findings to other populations. Additionally, patients with INOCA are a heterogeneous group and their prognoses may differ depending on the actual underlying mechanism of the acute presentation. Unfortunately, this is not routinely investigated in real-life settings, and our data also reflect this problem.

## Conclusions

The prognosis of patients with ACS appears to be related to their cardiovascular risk factors and baseline clinical presentation irrespective of whether significant atherosclerotic lesions are found and treated. The ACS incident should not be misjudged as a trivial event, even if cardiac biomarkers remain stable and no significant lesions are found on angiography. Optimal control for known cardiovascular risk factors should be an important part of the secondary prevention in patients with ACS INOCA. In future studies, more efforts should be made to identify the underlying mechanisms for their ACS presentations as this would allow for better tailored treatments and possibly improve their prognoses.

## Author Contributions

**Conceptualization:** Lukasz Zandecki, Zbigniew Siudak.

**Data curation:** Krzysztof Plens.

**Formal analysis:** Krzysztof Plens.

**Investigation:** Lukasz Zandecki, Agnieszka Janion-Sadowska, Jacek Kurzawski, Lukasz Piatek, Michal Zabojszcz, Marcin Sadowski.

**Methodology:** Krzysztof Plens, Zbigniew Siudak.

**Project administration:** Zbigniew Siudak.

**Resources:** Zbigniew Siudak.

**Supervision:** Zbigniew Siudak.

**Validation:** Lukasz Zandecki, Agnieszka Janion-Sadowska, Jacek Kurzawski, Lukasz Piatek, Michal Zabojszcz, Krzysztof Plens, Zbigniew Siudak, Marcin Sadowski.

**Writing – original draft:** Lukasz Zandecki.

**Writing – review & editing:** Lukasz Zandecki, Agnieszka Janion-Sadowska, Jacek Kurzawski, Lukasz Piatek, Michal Zabojszcz, Zbigniew Siudak, Marcin Sadowski.

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
