## [Decision Letter · Decision Letter 0]

9 Mar 2020

PONE-D-20-04772

Clinical presentation and 3-year outcomes of patients with acute coronary syndromes and non-obstructive coronary arteries on angiography

PLOS ONE

Dear Dr. Zandecki,

Thank you for submitting your manuscript to PLOS ONE. After careful consideration, we feel that it has merit but does not fully meet PLOS ONE’s publication criteria as it currently stands. Therefore, we invite you to submit a revised version of the manuscript that addresses the points raised during the review process.

We would appreciate receiving your revised manuscript by Apr 23 2020 11:59PM. To enhance the reproducibility of your results, we recommend that if applicable you deposit your laboratory protocols in protocols.io, where a protocol can be assigned its own identifier (DOI) such that it can be cited independently in the future. For instructions see: http://journals.plos.org/plosone/s/submission-guidelines#loc-laboratory-protocols

We look forward to receiving your revised manuscript.

Kind regards,

Corstiaan den Uil

Academic Editor

PLOS ONE

Journal Requirements:

Reviewers' comments:

Reviewer's Responses to Questions

**Comments to the Author**

1. Is the manuscript technically sound, and do the data support the conclusions?

Reviewer #1: Yes

Reviewer #2: Yes

2. Has the statistical analysis been performed appropriately and rigorously? 

Reviewer #1: Yes

Reviewer #2: Yes

3. Have the authors made all data underlying the findings in their manuscript fully available?

Reviewer #1: Yes

Reviewer #2: Yes

4. Is the manuscript presented in an intelligible fashion and written in standard English?

Reviewer #1: Yes

Reviewer #2: Yes

5. Review Comments to the Author

Reviewer #1: Excellent paper that compares ACS in two different presentations, with obstructive disease versus non obstructive disease and the prognosis implications. My only concern is not many data available about the underlying cause of MINOCA cases that need to be addressed (vasospasm, tako-tsubo, miocarditis....)

Reviewer #2: Overall:

- The authors should be complemented with their presenting work on INOCA ACS patients. They emphasize the importance of performing additional examination in this subset of ACS patients and that it should not be considered as a benign condition

- The authors should review their manuscript with regard to grammar and the use of punctuation by a native English speaker

- A few times, the abbreviation ACS INOCA is misspelled as ACS INCOA.

Introduction:

- The introduction is not logically structured since the authors start with MINOCA, then INOCA, then MINOCA and ending with INOCA. This is somewhat confusing for the reader.

- The terms MINOCA and INOCA are used interchangeably in the introduction. Although the term MINOCA and its pathophysiological mechanisms are discussed in dept, the authors include patients with INOCA in their study. The authors should update the introduction where they lie their focus more on INOCA.

Methods:

- Is ORPKI an abbreviation? Please write fully.

- Statistical analysis: the authors should exclude the web address (R).

Results:

- Where there patients excluded from analysis? If so, please present

- An additional table presenting outcome data of 36-month follow-up should be included in the manuscript.

- The demographic characteristics of the included patients are presented in the table but should me described in a comprehensive matter.

Tables:

- Percentages should be presented with only 1 decimal.

- Please write diabetes mellitus instead of diabetes

- Is the variable smoking current smoking or former smoking, or both?

- P values <0.001 are sometimes displayed as <0.001 and sometimes <.001.

- ICUS should be rewritten as IVUS.

- Table 2: Why is the composite endpoint MACE not presented in this table?

Figures:

- Poor quality

- Kaplan-Meier curve is misspelled as Kaplan-Maier.

Discussion:

- The authors should start this section what the main findings of this study were.

- Line 213-214: The comments “which make this group even more heterogenic” makes no sense in this context.

- The main problem (which already came forward in the introduction section) is that, although different, MINOCA and INOCA are sometimes used interchangeably and the data of the current study (i.e. INOCA) are directly compared to MINOCA data. Almost all relevant references are MINOCA papers. Probably it would be better to focus even more on INOCA, or perform analysis with ACS patients and elevated troponin (and thus try to analyse the MINOCA patients).

Conclusion:

- Should be described in a more comprehensive matter.

6. PLOS authors have the option to publish the peer review history of their article (what does this mean?). If published, this will include your full peer review and any attached files.

Reviewer #1: No

Reviewer #2: Yes: T.F.S. Pustjens

---

## [Author Response · Author response to Decision Letter 0]

23 May 2020

Reviewer #1: Excellent paper that compares ACS in two different presentations, with obstructive disease versus non obstructive disease and the prognosis implications. My only concern is not many data available about the underlying cause of MINOCA cases that need to be addressed (vasospasm, tako-tsubo, miocarditis....)

Thank you for your encouraging review and crucial comment about the underlying cause of MINOCA. Unfortunately, in most cases in our dataset, the underlying cause of (M)INOCA was not established. This study reflects the real-life situation in the region—and possibly in other regions in the world—where the explanation for the acute coronary presentation is not investigated further. We emphasized the importance of performing additional assessments in this subset of patients with ACS in the limitations and conclusions sections with a strong advice for physicians to add more efforts to understand the underlying cause of (M)INOCA. With such underutilization of additional tests, the prognoses of such patients are a concern. Maybe a better understanding of the underlying mechanisms in the acute presentation would allow for better tailored treatment and improve their prognoses.

 

Reviewer #2: Overall:

- The authors should be complemented with their presenting work on INOCA ACS patients. They emphasize the importance of performing additional examination in this subset of ACS patients and that it should not be considered as a benign condition

- The authors should review their manuscript with regard to grammar and the use of punctuation by a native English speaker

- A few times, the abbreviation ACS INOCA is misspelled as ACS INCOA.

The manuscript has been edited by a professional English editing service. We have attached the editing certificate as well. We have also corrected the misspelled abbreviations.

Introduction:

- The introduction is not logically structured since the authors start with MINOCA, then INOCA, then MINOCA and ending with INOCA. This is somewhat confusing for the reader.

- The terms MINOCA and INOCA are used interchangeably in the introduction. Although the term MINOCA and its pathophysiological mechanisms are discussed in dept, the authors include patients with INOCA in their study. The authors should update the introduction where they lie their focus more on INOCA.

We tried to distinguish the publications on MINOCA from those on INOCA by using the appropriate abbreviation when referring to the references. Our study focused on patients with ACS INOCA, which also includes those with MINOCA. However, most of the papers published so far are limited to MINOCA. The term “INOCA” has been introduced only recently, and this is one of the first papers to assess the prognosis in a broad group of patients who experienced ACS with non-obstructive coronary arteries. We realize that this may cause some confusion; therefore, we have edited some sections of the Introduction.

Methods:

- Is ORPKI an abbreviation? Please write fully.

Yes, this is an abbreviation. We have added both Polish and English expanded forms of the abbreviation in the manuscript.

- Statistical analysis: the authors should exclude the web address (R).

The suggested change has been incorporated in the revised version of the manuscript.

Results:

- Where there patients excluded from analysis? If so, please present

Overall, 27 patients were excluded from the analysis due to missing or inconsistent data. We have added this information in the manuscript.

- An additional table presenting outcome data of 36-month follow-up should be included in the manuscript.

The table has been added to the manuscript as suggested.

- The demographic characteristics of the included patients are presented in the table but should me described in a comprehensive matter.

We have added some text in the results section as suggested. 

Tables:

- Percentages should be presented with only 1 decimal.

- Please write diabetes mellitus instead of diabetes

The suggested change has been incorporated in the revised version of the manuscript.

- Is the variable smoking current smoking or former smoking, or both?

It refers to active smoking, and we have added this information to the manuscript. 

- P values <0.001 are sometimes displayed as <0.001 and sometimes <.001.

- ICUS should be rewritten as IVUS.

The suggested changes have been incorporated in the revised version of the manuscript.

- Table 2: Why is the composite endpoint MACE not presented in this table?

As suggested, we have added MACE to Table 2.

Figures:

- Poor quality

- Kaplan-Meier curve is misspelled as Kaplan-Maier.

We have improved the quality of the figures and corrected the misspelt words. 

Discussion:

- The authors should start this section what the main findings of this study were.

The suggested change has been incorporated in the revised version of the manuscript. 

- Line 213-214: The comments “which make this group even more heterogenic” makes no sense in this context.

This part of the sentence has been deleted.

- The main problem (which already came forward in the introduction section) is that, although different, MINOCA and INOCA are sometimes used interchangeably and the data of the current study (i.e. INOCA) are directly compared to MINOCA data. Almost all relevant references are MINOCA papers. Probably it would be better to focus even more on INOCA, or perform analysis with ACS patients and elevated troponin (and thus try to analyse the MINOCA patients).

Most of the current papers focus on MINOCA. The concept of INOCA is less popular, although not less important in our opinion. The current study aimed to focus on this subset of patients with ACS who underwent urgent coronary angiography due to suspected ACS irrespective of biomarkers. All those patients were collectively analyzed and termed “ACS INOCA” following the nomenclature proposed in previous papers. Comparing MINOCA with non-MINOCA based on our available data would be of limited value because MINOCA was considerably less frequent.

Conclusion:

- Should be described in a more comprehensive matter. 

Thank you for pointing out this important aspect. In accordance with your comment, the section has been revised in the revised version of the manuscript.

---

## [Editor Report · Decision Letter 1]

2 Jun 2020

Clinical presentation and 3-year outcomes of patients with acute coronary syndromes and non-obstructive coronary arteries on angiography

PONE-D-20-04772R1

Dear Dr. Zandecki,

We are pleased to inform you that your manuscript has been judged scientifically suitable for publication and will be formally accepted for publication once it complies with all outstanding technical requirements.

With kind regards,

Corstiaan den Uil

Academic Editor

PLOS ONE
---

## [Editor Report · Acceptance letter]

5 Jun 2020

PONE-D-20-04772R1 

Clinical presentation and 3-year outcomes of patients with acute coronary syndromes and non-obstructive coronary arteries on angiography 

Dear Dr. Zandecki:

I'm pleased to inform you that your manuscript has been deemed suitable for publication in PLOS ONE. Congratulations! Your manuscript is now with our production department. 

Kind regards, 

on behalf of

Dr. Corstiaan den Uil 

Academic Editor

PLOS ONE